# Particulate Matter in Human Elderly: Higher Susceptibility to Cognitive Decline and Age-Related Diseases

**DOI:** 10.3390/biom14010035

**Published:** 2023-12-26

**Authors:** Caridad López-Granero, Leona Polyanskaya, Diego Ruiz-Sobremazas, Angel Barrasa, Michael Aschner, Matilde Alique

**Affiliations:** 1Department of Psychology and Sociology, University of Zaragoza, 44003 Teruel, Spain; cgranero@unizar.es (C.L.-G.); d.ruiz@unizar.es (D.R.-S.); abarrasa@unizar.es (A.B.); 2Faculty of Psychology and Educational Sciences, University of Coimbra, 3000-115 Coimbra, Portugal; leona.polyanskaya@uc.pt; 3Coimbra Institute for Biomedical Imaging and Translational Research, University of Coimbra, 3000-548 Coimbra, Portugal; 4Department of Molecular Pharmacology, Albert Einstein College of Medicine, Bronx, NY 10461, USA; michael.aschner@einsteinmed.edu; 5Departamento de Biología de Sistemas, Universidad de Alcalá, Alcalá de Henares, 28871 Madrid, Spain; 6Instituto Ramón y Cajal de Investigación Sanitaria (IRYCIS), 28034 Madrid, Spain

**Keywords:** particulate matter, air quality, elderly, cognitive function, neurodegeneration, extracellular vesicles

## Abstract

This review highlights the significant impact of air quality, specifically particulate matter (PM), on cognitive decline and age-related diseases in the elderly. Despite established links to other pathologies, such as respiratory and cardiovascular illnesses, there is a pressing need for increased attention to the association between air pollution and cognitive aging, given the rising prevalence of neurocognitive disorders. PM sources are from diverse origins, including industrial activities and combustion engines, categorized into PM_10_, PM_2.5_, and ultrafine PM (UFPM), and emphasized health risks from both outdoor and indoor exposure. Long-term PM exposure, notably PM_2.5_, has correlated with declines in cognitive function, with a specific vulnerability observed in women. Recently, extracellular vesicles (EVs) have been explored due to the interplay between them, PM exposure, and human aging, highlighting the crucial role of EVs, especially exosomes, in mediating the complex relationship between PM exposure and chronic diseases, particularly neurological disorders. To sum up, we have compiled the pieces of evidence that show the potential contribution of PM exposure to cognitive aging and the role of EVs in mediating PM-induced cognitive impairment, which presents a promising avenue for future research and development of therapeutic strategies. Finally, this review emphasizes the need for policy changes and increased public awareness to mitigate air pollution, especially among vulnerable populations such as the elderly.

## 1. Significance: Why This Review?

The United States Environmental Protection Agency (EPA) recommends the use of Air Quality Index (AQI) forecasts to plan for outdoor life activities [1]. Hence, a suitable day for outdoor activity depends on the AQI, with values between 0 and 50 considered optimal and values between 101 and 300 considered unhealthy for human health. This proposal from EPA reflects the premise that our actions and health are subject to air quality and, therefore, by where we reside. Thus, poor indoor and outdoor air pollution and its effects on population health are one of the most important environmental and public health issues [2].

In 2014, the World Health Organization (WHO) warned about yearly seven million premature deaths associated with air pollution [3]. In a recent meta-analysis review, other authors identified 14 air pollutants associated with different diseases and mortality diagnoses, indicating that particulate matter (PM) as an air pollutant has grave effects on human health [4].

Several publications have shown that air pollution is associated with an increased risk of respiratory, cardiovascular, and cerebrovascular disease [3], but its association with cognitive functioning and risk for older age people needs greater attention. It is widely accepted that aging is associated with the deterioration of physical and mental capabilities, often resulting in neurocognitive disorders, such as dementia [4]. Although cognitive decline is related to increased age [5], age is not a manipulated factor. Thus, with the increased prevalence of neurocognitive disorders, efforts should be centered on defining other factors implicated in the aging process that could be susceptible to manipulation or variation [6,7,8]. If later-life cognitive decline is not necessarily associated with typical Alzheimer’s disease pathology, could there be other important environmental determinants of cognitive decline? Could PM exposure be an important contributor to cognitive aging? Could PM trigger neuroinflammatory processes and extracellular vesicles (EVs) alterations that contribute to neurological diseases?

Our research group has previously addressed the influence of air pollution in developmental animal models [9,10]. Given the impact that air pollution, especially PM, might have on the aging process and its propensity to hasten neurodegeneration, the present review focuses on understanding inflammatory-dependent aging processes. We highlight the repercussions of poor air quality on the nervous system in terms of cognitive decline during aging, its relationship to inflammatory responses, and the role of EVs.

## 2. Etiology and Route of Access of Particulate Matter to the Human Body

The primary origin of PM is from a combination of industrial activity, internal combustion engines, and geographic and meteorological conditions linked to the increased forest fires [11]. Specifically, smoke originating from forest fires is primarily responsible for ultrafine particles (UFPM ≤ 100 nm in diameter) [11]. Inspiring such particles increases the risk of developing respiratory diseases, such as tonsillitis, asthma, bronchitis, lung cancer, and cystic fibrosis pneumonia later in life [12]. Although the impact of air quality on respiratory health has become more pressing, in this review, we emphasize the need to focus on cognitive function, as it has been observed that PM readily accumulates in the brain with the potential to induce neurodegeneration [11,13].

### 2.1. Etiology of Particulate Matter

Air quality is determined by estimating the number of solid matter particles in the air [2]. These particles contribute to the air pollution level and are referred to as PM [14]. PM are partly formed in the atmosphere through chemical reactions that produce inorganic nitrates and sulfates, as well as organic compounds [2]. Inhalable particles are divided into four groups by diameter highlighting three of them: larger than 10 μm (μm), smaller than 10 μm (PM_10_), and smaller than 2.5 μm (PM_2.5_) [15]. In addition, a fourth group exists, referred to as ultrafine PM (UFPM or PM_0.1_, with diameter < 100 nm) [16]. PM_2.5_ are not visible by the naked eye, and levels of such particles (measured in μg/m^3^) are not distinguished without specialized equipment when kept at concentrations below 35 μg/m^3^ [17]. The US EPA, the European Environment Agency (EEA), and the WHO set a PM_2.5_ safety standard under 12 μg/m^3^ (annual average) and a 24 h average of 35 μg/m^3^. Nonetheless, this value is routinely exceeded worldwide (e.g., in large cities in India, China, or Mexico) [16]. Therefore, many countries have failed to keep urban PM pollution levels within WHO guidelines.

PM is mainly derived from road transport, agriculture, power plants, industry, and homes. Outdoor particles are primarily produced from traffic-related activities [16]. PM_2.5_ is a mixture of harmful chemicals, such as sulfur dioxide, nitrogen dioxide, organic phosphates, and chlorine, which are utilized in agriculture, bisphenol, etc., along with combustion particles, carbon particles, heavy metals, and pollutants from industrial sites and factories. In contrast, PM_10_ consists mainly of pollen, mold, and dust particles [18,19]. Hence, despite being less visible to the public, <PM_2.5_ poses much greater health risks than larger particles, and due to their size, with limited efficacy of protective gear, such as masks or respirators, in preventing exposure [2]. UFPM are mostly derived from wildfire smoke [11] (see Table 1). Indoor environmental particles are derived from combustion activities such as cooking, as well as heating with coal, wood or dung, candles, incense, kerosene lamps, and tobacco smoking. In addition, indoor pollution originates from non-combustion sources and volatile organic compounds such as cleaning and insecticide products, electric devices, and printers [2].

It is noteworthy that PM pollution also has important economic ramifications since it increases medical expenses, reduces worker productivity, and damages soil, crops, forests, lakes, and rivers [2].

### 2.2. Route of Access and Biological Systems of Particulate Matter

PM can reach and accumulate throughout tissues and organs in humans, such as the gastrointestinal tract, skin, mucosae, placenta, and brain [11].

The cardiovascular system is affected by PM exposure [16,20]. The probability of heart attack due to coronary ischemia, coronary revascularization, ischemic heart disease, thrombosis, or stroke is also positively correlated with air pollution [21,22], especially among the older population [23,24,25], with evidence indicating a possible biological sex bias, where women are affected to a greater extent than men [24,25]. Notably, non-industrial fine airborne particles (e.g., dust particles from dry desert areas) contribute to the probability of developing cardiovascular diseases as much as anthropogenic PM_2.5_, e.g., emitted from industrial (e.g., metals) and agricultural (organic chemicals) sites [26]. While the impact of air pollution on respiratory health can manifest even after brief exposure to elevated levels of PM_2.5_, the effects on the cardiovascular system are best observed through longitudinal studies comparing populations exposed to long-term high levels of PM_2.5_ to those living in areas with PM_2.5_ levels within the safety limit of 10–12 μg/m^3^ [27].

The brain is especially vulnerable to air pollution. Inhalation is the main route of PM exposure [16,28,29]. Costa and collaborators characterized PM absorption into the brain [16]. PM enters the central nervous system (CNS) through the nose but follows different paths depending on the size and solubility. While PM_10_ is deposited in the upper airways, PM_2.5_ is deposited in the lungs and is also subject to olfactory transport and deposition in the olfactory cortex and other brain regions [16,29,30]. Furthermore, UFPM can be deposited in the cerebral cortex and cerebellum secondary to transport via the olfactory nerves [28]. Once reaching the lung, PM may also travel through the blood into the brain [16,30] (See Table 1). Understanding these pathways and how PM is capable of reaching the CNS underscores the urgent need for an in-depth analysis of its impact on cognitive function. In the brain, PM accumulation contributes to CNS diseases [12,13,31]. The impact on older people is such that there is even evidence of a greater accumulation of amyloid beta (a characteristic of Alzheimer’s disease, AD) in individuals exposed to higher levels of air pollution [13].

Overall, the decrease in PM_2.5_ concentration correlates with the reduction of hospitalization and early death rates [32], decreasing the burden on economic sectors and the health care system. Altogether, PM exposure affects the respiratory, cardiovascular, and autoimmune systems, potentially accelerating its adverse effects on cognitive functions (Figure 1).
biomolecules-14-00035-t001_Table 1Table 1Summaries of etiology of particulate matter and pathways implicated until reaching the CNS in the brain.PM by Diameter^1^ Main EtiologyDeposited in:ReferencesPM_0.1_ or UFPM≤100 nmSmoke from wildfireCerebral cortex and cerebellum secondary to transport via the olfactory nerves[11,28]PM_2.5_≤2.5 μmTraffic-related activities, industrial sites, factories, and agriculture^2^ Lungs and is also subject to olfactory transport and deposition in the olfactory cortex and other brain regions[16,18,19,30]PM_10_≤10 μmPollen, mold, and dust particlesFiltered out by thenose and upper airways[16,18,19,29,30]UFPM: ultrafine particles; CNS: central nervous system. Notes: ^1^ Outdoor environmental particles are primarily produced from road transport, agriculture, power plants, industry, and forest fires [11,16]. Indoor environmental particles are derived from combustion activities such as cooking, as well as and heating with coal, wood or dung, candles, incense, kerosene lamps, tobacco smoking, non-combustion sources, and volatile organic compounds such as cleaning and insecticide products, electric devices, and printers [2]. ^2^ PM enters through the nose; once the lungs are reached, PM may also travel through the blood into the brain [16,30].

## 3. Role of Particulate Matter in Cognitive Functions in Human Elderly

Air pollution levels negatively correlate with working memory span, error monitoring, attention, and behavioral control both in children [33,34,35,36] and the elderly [37,38]. Several authors have even indicated a 2-year excess decline in cognitive function in relation to each 10 mg/m^3^ increase in long-term PM exposure in older women [39] (see Table 2). The decline in cognitive function in later life may even result in neurodegenerative disorders and dementia at later life stages [11,38,40,41].

### 3.1. Cognitive Functions Affected

Table 2 summarizes the relationship between PM and cognitive function. Several studies have shown significant cognitive declines in relation to long-term exposure to indices of urban pollution [39,42,43]. A few epidemiological studies have demonstrated that even brief exposure to elevated concentrations of PM_2.5_, whether from indoor sources such as burning candles, which are a significant source of PM_2.5_, or outdoor sources such as commuting on a busy road, may cause a rapid decline in cognitive function, as demonstrated by various cognitive tests assessing visual–spatial processing, executive function, verbal fluency, memory, attention, and orientation [41,44,45,46]. The positive correlation between PM_2.5_ concentration and cognitive performance is robust in population cohorts between 65 and 79 years [47]. This study indicates that cognitive performance in men is more susceptible to the adverse effects of air pollution than in women [47].

However, the literature is in disagreement regarding the relationship between sex and air pollution. Several authors indicated that exposure to PM_1_, PM_2.5_, and PM_10_ affects cognition to a greater extent in older women than men [39,44]. Specifically, decreased levels of episodic memory and mental status among middle-aged and elderly Chinese were noted. In addition, they found significantly worse effects of PM_1_ and PM_2.5_ on cognitive function in women than men. Similarly, in the United States, a study demonstrated greater cognitive decline in older women [39]. In addition, a new study has indicated that PM_2.5_ exposure may accelerate cognitive aging in middle-aged and older adults, suggesting significant sex disparity with higher vulnerability in women [48]. In this study, women displayed worse performance in global cognitive scores, summing up response scores on immediate word recall, delayed word recall (both are identified as the domain of episodic memory), orientation, visuo-construction, and numeric ability (mental status).

In addition, the association between long-term PM exposure and reasoning, memory, and phonemic and semantic fluency functions in people’s mean age was addressed in 66 year-old individuals residing in London [45]. The authors observed lower scores in reasoning and memory tests in individuals exposed to PM but not in verbal fluency. In addition, the authors compared this link between cognitive decline and PM exposure across particle sources. They concluded that traffic-related particles were not more strongly associated with cognitive function compared with particles from all sources combined [45]. In addition, a 2.97 μg/m^3^ increase in PM_2.5_ exposure was related to poor memory and executive function. Furthermore, per 2.05 μg/m^3^ increase in PM_2.5–10_ exposure, global cognition, attention, and verbal fluency declined. Also, per 4.94 μg/m^3^ increase in PM_10_ exposure, poor global cognition was noted in older adults [49]. The authors indicated that PM_2.5_, PM_2.5–10_, and PM_10_ corresponded to 1.4, 5.8, and 2.8 years of aging, respectively. Additionally, studies observed that higher exposures to PM_10_, PM_2.5_, and PM _2.5abs_ (0.0001/m) were associated with a decline in the immediate verbal memory test in adults between 45 and 75 years old [50].

Episodic memory is repeatedly affected by PM exposure [39,40,44]. In preclinical stages, episodic memory was a critical factor in the etiology of AD [51]. Younan and collaborators, in 2020, observed an association between PM exposure, episodic memory affectation (immediate recall and new learning), and dementia risk [40]. These authors found that the decline in cognitive function associated with PM_2.5_ could be mediated by AD pattern similarity scores by the observation of grey matter atrophies in the brain, as verified by magnetic resonance imaging (MRI).

### 3.2. Cognitive Reserve Concept

Although age-related morphological changes in the CNS are inevitable, individuals may differ regarding how they compensate for them [52]. Efforts have been focused on understanding why some individuals regain more functionality and recover faster after stroke or brain surgery. Others need more time and may not reach the pre-lesion state, while some do not recover at all [53]. The ability of the brain to resist age-related degenerative changes or disease-related neurological damage is often referred to as cognitive reserve [54,55].

Individuals develop various cognitive strategies to compensate for the age- or disease-related functional loss in the neural substrates or, due to neural plasticity, develop differential neural networks recruited for certain cognitive tasks to compensate for the age- or disease-related loss of neural substrates [56]. The concept of cognitive reserve as a unifying construct employs compensatory mechanisms, which extend the effects of given cognitive damage [57,58,59]. Thus, those who display a greater cognitive reserve will have more success when faced with pathological aging or will be more likely to age in a more satisfactory manner [56,57,59].

Although some factors contributing to the cognitive reserve are purely genetic, anatomical, and morphological, a broad range of factors is either epigenetic (e.g., environmental during fetus development) or ontogenetic (environmental exposures during individual development, physical activity, level of education and occupation, lifestyle, psychological stress, etc.) [56]. As PM_2.5_ concentration is related to many of these factors, air quality could play an important role in the ability of the cognitive system to resist the adverse effects associated with aging.

The literature has yet to address human cognitive reserve in relation to air pollution. Nonetheless, indirectly, air pollution could affect the cognitive reserve by increasing the level of psychological distress [60,61,62] or by impacting morphological changes in the elderly brain [63,64,65], as well as by potentially reducing the ability to re-route neural signals and recruit differential neural networks for performing cognitive tasks in case of neuronal age-related neuron losses [66,67]. Hence, by negatively affecting morphological changes in the brain, air pollution may reduce the ability to re-route neural signals and recruit differential neural networks for performing cognitive tasks in case of neuronal age-related neuron losses.
biomolecules-14-00035-t002_Table 2Table 2Summaries of particulate matter exposure and cognitive functions altered by PM exposure in older people.ReferenceAuthor’s Name/YearCountryPM ExposureCognitive Function AffectedSex-Dependant?[39]Weuve et al., 2012USAPM_2.5–10_, PM_2.5_GCF, verbal memory, digit span, and verbal fluency scores↓ in women[40]Younan et al., 2020USAPM_2.5_Episodic memory-[44]Yao et al., 2022ChinaPM_1_, PM_2.5_, PM_10_Episodic memory, mental status↓ in women[45]Tonne et al., 2014UKPM_2.5_, PM_10_Reasoning and memory-[46]Wurth et al., 2018Puerto Rican in USAPM_2.5_RecognitionNo[47]Wang et al., 2020ChinaPM_2.5_MMSE↓ in men[48]Mo et al., 2023ChinaPM_2.5_GCF↓ in women[50]Ogurtsova et al., 2023GermanyPM_10_, PM_2.5_
Immediate verbal memoryNoNote: ↓: poor performance; GCF: Global Cognitive Function; -: Not analyzed; MMSE: Mini-Mental State Examination.

## 4. Particulate Matter and Neurodegenerative Diseases in Human Elderly

The relationship between neurodegeneration and aging can be affected by PM composition, size, and adsorption [68]. Independent of these three aspects, PM has been associated with different diseases, including those of the CNS [69]. This relationship, alongside the combination of PM_10_ emitted to the atmosphere by industrial processes (47.4%) and stationary fuel combustion (39%) and a growing elderly population, may create a favorable condition, accounting for the increment of different neurodegenerative diseases in the elderly [70].

Neurodegenerative etiology is highly complicated; however, there are some common points between the most frequent neurodegenerative diseases, oxidative stress [71], and neuroinflammation [72]. There is no clear mechanism by which PM impacts the nervous system, but the most accepted theory is inflammation [73]. Peripheral inflammation can affect the CNS’s homeostasis by activating microglia and inducing neuroinflammation with an abnormal protein aggregation [74].

### 4.1. Inflammatory Processes

Air quality is involved in physiological mechanisms that can directly affect cognition [37,38] by causing neuroinflammation and oxidative stress [31]. Several studies have shown that air pollutants can readily cross the blood–brain barrier, concomitantly causing systemic inflammation that adversely impacts the developing nervous system [75]. Specifically, the neuroinflammatory response is recognized as a causal factor in CNS diseases [31]. Pro-inflammatory molecules were detected in blood as well as in cerebrospinal fluid after air pollution exposure [76]. The presence of these molecules causes brain damage by reducing neural tissue in specific brain areas and by increasing the blood–brain barrier permeability [77].

PM can induce a variety of cellular, molecular, and inflammatory pathways which generate brain damage. One neuroinflammatory process is microglial activation, resulting in the synthesis and release of pro-inflammatory molecules. When microglia are overstimulated, an increase of tumor necrosis factor alpha (TNF-α), interleukin-1 beta (IL-1β), type II interferon (IFN-γ), and reactive oxygen species (ROS) production takes place, with associated neurotoxic effects [78]. However, there is insufficient information on neuroinflammation derived from air pollution in elderly people. This relationship has been studied only in children, youths, adults, and animal models.

Calderón-Garcidueñas, in 2004 [13], conducted a study using brain samples from Mexico City and Monterrey subjects whose cause of death was not an infection, inflammatory events, brain ischemia, or hypoxia. Higher levels of pro-inflammatory markers (cyclooxygenase-2, COX2) were found in the frontal cortex and hippocampus of individuals exposed to higher levels of air pollution. Furthermore, this relationship was only maintained in the highest exposed group, being non-significant in the lowest exposed subjects. In addition, increased levels of beta amyloid 42 (Aβ-42) levels were detected in the same areas where the pro-inflammatory markers were present. Another study with brain samples from Mexico City found differences in *COX2* mRNA expression in the frontal cortex and substantia nigra, absent any differences in the hippocampus. Subjects carrying human apolipoprotein E (APOE) 3/3 had the highest *COX2* expression. Aβ-42 was also identified in the olfactory nerve of subjects with APOE 3/3 [79]. Neuroinflammation and Aβ-42 aggregation were identified in the brains of people living in highly polluted cities at much earlier stages than expected. Thus, while changes are clearly associated with normal aging, air pollution hastens the normal aging process [68].

### 4.2. Mitochondrial Function

Few studies have shown that mitochondria are targets in PM-induced human neurodegenerative disorders. PM can induce oxidative stress, and mitochondrial DNA has been shown to be affected due to altered DNA methylation; therefore, mitochondrial damage [80] secondary to air PM pollution is associated with the development of human cardiovascular and neurodegenerative diseases. Furthermore, PM may lead to dysregulation of neurohormonal stress pathways and trigger inflammation as well as oxidative stress affecting mitochondria. The mechanistic impact of PM-induced mitochondrial damage on human neurodegenerative diseases is incompletely understood [81]. Therefore, further investigations on mitochondrial dysfunction associated with human neurodegenerative diseases by PM require an understanding of this complex pathogenesis.

### 4.3. Changes in Gene Expressions

The impact of air exposure has been linked to the development of neurodegenerative diseases, such as PD, associated with genetic alterations [82]. The molecular mechanisms of PD pathogenesis can be associated with epigenetic changes that alter gene expression without affecting the DNA sequence through DNA methylation, histone post-transcriptional modifications, and non-coding RNAs. However, environmental exposure can also cause alterations in cell function owing to changes in gene expression [82]. In PD, the most important epigenetic modifications are as follows: (1) DNA methylation (SNCA, PGC-1α, TNF- α, NOS2), (2) post-transcriptional histone methylation (H3K14, H3K9, and H3K18), and (3) microRNAs and other non-coding RNAs alterations [83,84]. Moreover, DNA methylation is related to other neurodegenerative disorders such as AD, depression, and Rett Syndrome. The most important genes for DNA methylation in AD are APP, BACE1, BIN1, and ANK1, as well as the co-occurring modifications to histones and expression of non-coding RNAs [85]. Furthermore, it is well established that PM has pro-oxidant activity, damaging mitochondrial DNA that is more susceptible to oxidative injury, and finally, mitochondrial DNA becomes methylated upon air pollution [80]. Lastly, an in vitro study in primary human neurons exposure to ultrafine PM has shown changes in non-coding RNAs implied in epigenetic regulation of gene expression and, therefore, alterations in gene expression related to neuroprotection such as a significant increase in the expression of metallothionein 1A and 1F with the propose to reduce PM damage in the brain [86].

### 4.4. Neurodegenerative Diseases

The first association between air pollutants and cognitive impairment was reported in 2000 [87]. Since then, the importance of air pollution and cognition has continuously risen. This initial relationship was established between participants living near a highway or a major roadway. Most of the published studies have attempted to explain the relationship between PM exposure and neurodevelopmental diseases using preclinical models; however, few articles have disentangled this relationship in humans.

In the United States, it is estimated that 29 million and 88 million people are exposed to PM_10_ and PM_2.5_, respectively [88]. Recently, air pollution has been linked to neurodegenerative disorders like Alzheimer’s disease (AD) and Parkinson’s disease (PD) [41]. Jung and collaborators found a 138% increase in the risk of developing AD per increase of 4.34 μg/m^3^ in PM_2.5_ [89]. This study was performed on people aged 65 years of age or older in Taiwan. However, no data regarding cognitive performance were shown. As noted before, other work had shown that long-term exposure to PM_10_ and PM_2.5_ was linked to cognitive dysfunction, mainly affecting episodic memory [39,42,90,91] (Telephone Interview for cognition; cognition tests (Weschler Adult Intelligence Scale Revised; Mini-mental; CERAD Test; Boston memory test, among others). These domains are critical for the neurodegenerative process, especially for AD and PD.

Other studies found a 1.17 increased risk of developing AD after exposure to PM_2.5_ [92]. This increased risk for AD after PM_2.5_ exposure was also seen in the Women’s Health Initiative, which found that AD diagnosis was 1.28 more probable in black than white women.

Few studies have controlled pollution exposure sources. Specifically, wood burning and traffic exhaust of PM_2.5_ were related to increased dementia risk [92,93,94]. These studies used data from the Betula database, finding that when the PM_2.5_ from residential wood burning and traffic exhaust was combined with other variables (smoking, body-mass-index, waist/hip ratio, alcohol consumption, among others) controlled, the probability of developing dementia was 1.55 more frequent per 1 µg/m^3^. If PM_2.5_ is considered alone, the source increasing the probability of dementia most was traffic exhaust of PM_2.5_, with a Hazard Ratio of 1.71 per 1 µg/m^3^ [95]. One of the key findings in this study was the number of variables that may impact the relationship between air pollution and neurodegenerative diseases, such as health style, genetic conditions, and environmental conditions.

Another cohort of subjects mined was the Adult Changes in Thought (ACT) from a Washington State population aged 65 or more [96]. The hazard of AD diagnosis was 11% greater after 1 µg/m^3^ increase in ten years of average PM_2.5_. However, the results were inconsistent, showing even an inverse association between PM_2.5_ and dementia diagnosis. In contrast to the Betula database’s studies, health style variables were not related to dementia diagnoses [96]. It is noteworthy that there is no magnetic resonance imaging data on PM_2.5_ in elderly people. Evidence in youths showed reduced white matter and subcortical and cortical areas in adolescents/children [97]. Furthermore, a linear decline in episodic memory (a key symptom in AD) was present related to PM_2.5_ [40].

Recent meta-analyses show that PM_2.5_ is also related to PD risk; however, this effect should be carefully considered due to the effect size and variation that can be detected in the studies included [98]. Several articles attempted to relate PM_2.5_ with PD by analyzing hospital administrations in different cities and finding a positive relationship between AD and PD [99]. Nevertheless, other studies failed to support this relationship either with men [100] or women [101]. One of the main differences between these studies is how the subjects were recruited. Kioumourtzoglou’s study [99] used subjects on first hospital admission for dementia, AD, or PD. At the same time, Palacios and collaborators [100,101] used a large prospective cohort of subjects.

In addition, the relationship between air pollution and any type of dementia might be sex-dependent. Liu and colleagues found a higher odds ratio (OR) in females than males (PM_2.5′s_ OR = 1.29, PM_10′_s OR = 1.65; PM_2.5_ and PM_10′_s OR = 1.02) [102]. This work controls multiple variables while using a large sample size of people suffering from PD in the United States. However, the exposure time analyzed for air pollution was short. In contrast, other authors found increased OR on PM_10_ exposure, analyzing 1 to 18 years of exposure in a very large population of PD diagnosed [103]. Even though they assessed other molecules related to air pollution, they did not provide data regarding PM_2.5_.

#### Other Neurodegenerative Diseases

There are few published studies on the relationship between PM and other neurodegenerative diseases like sclerosis (multiple or Amyotrophic Lateral Sclerosis [ALS]) or other types of dementia such as frontotemporal. Regarding multiple sclerosis, some authors have found no increased probability of developing multiple sclerosis after PM exposure [104]. On the contrary, in another study using 9,072,576 participants, it was noted that PM_10_ concentrations explain 26.2% of the variance [105]. However, the multiple sclerosis cases in this article were self-reported, with imprecise air pollution measurements. For ALS, there is more evidence of this relationship. On the one hand, some works show no evidence of an increment in ALS risk after PM exposure [104,106,107]. On the other hand, other authors pointed out a slight increment in ALS prevalence [108] or visits to emergency departments for ALS symptoms [109]. In the case of other types of dementia, the published literature has not indicated an association linking PM_2.5_ and frontotemporal dementia [92].

Taken together, scientific evidence shows that air pollutants, specifically PM_10_ and PM_2.5_, are related to neurodegenerative disorders like AD and PD. However, there is still insufficient information about the variables that may impact that relationship or the link between PM and other neurodegenerative diseases other than AD and PD. Further research is needed to fully understand how the social environment, lifestyle, and other sociodemographic variables affect the relationship between air pollution and neurodegeneration.

## 5. Particulate Matter, Extracellular Vesicles and Human Aging

EVs are membranous particles with diameters ranging from 30 nm to 5 µm that are released by virtually all cells. Initially, EVs were thought only to remove unwanted cellular contents; however, more recently Mas-Margues and collaborators (2023) called EVs “very important particles” (VIPs) due to their crucial role in cellular communication, as they transport bioactive proteins, lipids, and nucleic acids as part of their functional cargo [110]. In addition, EVs serve as regulators of signaling processes, play significant roles in aging and age-related diseases, and have been used as a non-invasive approach for identifying new biomarkers of numerous illnesses [110].

### 5.1. Extracellular Vesicles, Particulate Matter and Age-Related Diseases

To date, several studies have demonstrated that EVs are released in response to environmental stimuli as potential mediators of the effects of PM exposure. EVs act as indicators of aging-associated human diseases in response to stress, including PM exposure, a highlighted condition that occurs with premature aging, including chronic disorders [111,112,113]. In addition, EVs released upon PM exposure provoke disease outcomes in the molecular mechanism. The INSIDE study identified the health impacts of PM exposure during pregnancy and hypertensive disorder development, demonstrating a link between PM exposure and EVs, such as biomarkers and mediators of cardiovascular disease [114]. Moreover, another study demonstrated that PM_2.5_ is associated with endothelial dysfunction and increases cardiovascular morbidity and mortality risk due to vascular aging. In this way, PM_2.5_ exposure was associated with elevated levels of endothelial EVs, proinflammatory cytokines, and proangiogenic factors [115].

More recently, respiratory exposures such as cigarette smoke and air pollution such as PM have been shown to lead to EV release, contributing to chronic diseases (cardiovascular disease, chronic obstructive pulmonary disease, lung cancer, and allergic asthma), and leading to a pro-inflammatory and pro-thrombotic environment that increases the risk of related premature aging diseases. These studies showed that EVs modulate several pathophysiological processes, such as inflammation, thrombosis, endothelial dysfunction, tissue remodeling, and angiogenesis, which increase the risk of related diseases [113,116]. While there is evidence of elevated concentrations of circulating endothelial-derived EVs in response to respiratory toxicants [117], further research is needed to fully understand their role in these disease etiologies, including the biological action of EV cargo and determinants of release, as well as their potential targets for preventing (biomarkers) and treating (therapeutic strategies) pollutant-related diseases. Therefore, the link between environmental pollutants and EVs in developing chronic systemic disorders is still under study.

### 5.2. Extracellular Vesicles, Particulate Matter, and Neurological Diseases

Knowledge of PM-related EV release associated with neurological diseases is scarce. As noted earlier, a few studies have demonstrated that chronic exposure to environmental toxins can cause various adverse health effects, including neurological disorders and carcinomas. A type of EVs, namely exosomes, has a cell-specific cargo that can change the fate of recipient cells and influence distantly located cells and tissues [104], playing an active role in intercellular signaling. Harischandra and colleagues showed that environmental toxins (such as PM) stimulate exosome release, increasing the risk for progressive neurodegenerative diseases, such as AD, PD, and Huntington’s disease, as well as certain cancers [118]. Recent work also showed a relationship between air pollution exposure and cognitive impairment and neurological disorders, focusing on EVs as critical signals from brain cells. Moreover, the potential use of neural-derived EVs as diagnostic or therapeutic molecules in air pollution-related cognitive impairment and neurodegeneration plays an essential role in fully understanding their contribution to PM-related neurological disorders [119].

In this regard, a recent review has shown that EVs may transport toxic forms of amyloidogenic proteins in neurodegenerative diseases and spread them to recipient cells in the CNS. EVs from the CNS can readily cross the blood–brain barrier and can be found in other body fluids, making them an attractive source as biomarkers of Parkinson’s disease and other neurodegenerative disorders [120].

In summary, EVs play a critical role in the response to environmental exposure (Figure 2); however, further research is needed to fully understand their contribution to neurological disorders associated with air pollution exposure (PM).

## 6. Conclusions and Future Perspectives

Despite increasing interest in the study of the consequences of air pollution exposure in humans, a scant number of studies addressed the effects of poor air quality on brain and CNS diseases. Several publications indicated that air pollution is associated with an increased risk of respiratory, cardiovascular, and cerebrovascular disease, but its association with cognitive functioning and impairment needs more specific and careful characterization, as does its relationship with age-related diseases. Most publications to date have addressed the effects of PM in young people, bypassing other large population risk groups, such as older age people. Therefore, as highlighted in this review, an urgent need exists to deepen our understanding of cognitive-dependent aging alterations and age-related disorders associated with PM exposures.

Could PM exposure be an important contributor to cognitive aging? The literature indicates that exposure to PM is capable of reaching and accumulating in almost all tissues and organs in humans. The impact of PM on cognitive function is well documented, showing that poor air quality is associated with a 2-year excess decline in cognitive function in relation to each 10 mg/m^3^ increase in long-term PM exposure [39], concomitant with effects on visual–spatial processing, executive function, verbal fluency, memory, attention, and orientation cognitive functions during elderly [41,44,45,46]. Additionally, women appear to be more susceptible to the adverse effects of PM exposure compared to men [39,44,48]. Altered cognitive ability might represent the anteroom of developing neurodegenerative disorders secondary to inflammatory alterations. Therefore, EVs could play an essential role in cognitive impairment after PM exposure by modulating neuroinflammatory processes. Indeed, EVs have been associated with chronic diseases, contributing to age-related diseases. A better understanding of EVs, the inflammatory system, and cognitive decline upon exposure to PM pollution is urgently needed.

Finally, in line with Schraufnagel and collaborators [121], it is noteworthy that air pollution reduction is possible, and the benefits of actions directed at such policies can result in substantial health gains, especially in risk populations such as elderly people. Several authors have even indicated that air pollution policies not only can improve human health but are also relevant for decreasing climate change. For example, Slovic and collaborators addressed how urban policies may improve air quality and mitigate global climate change [122]. Local initiatives and public awareness have largely focused on fuels and technology [122]. The benefits of the substitution of diesel taxis for alternative fuels have also been addressed (such as hybrid, natural gas, and liquefied petroleum gases) [123]. In Madrid, Spain, Vedrenne and collaborators examined the impact of Life Cycle Assessment (LCA) as an instrument for policy support [123], a means for studying the environmental impact of transport modes from cradle to grave [124]. The results showed that the shift to ecologic alternatives has a positive impact on climate change and air pollution. Undoubtedly, any environmental policy will improve air quality, which translates into a healthier situation for the elderly living in cities.

Thus, changes in local policy-setting agencies and governments, along with greater awareness on the part of the general population, will be the key to imparting a healthier natural environment and reduced exposure to air contaminants.

## Figures and Tables

**Figure 1 biomolecules-14-00035-f001:**
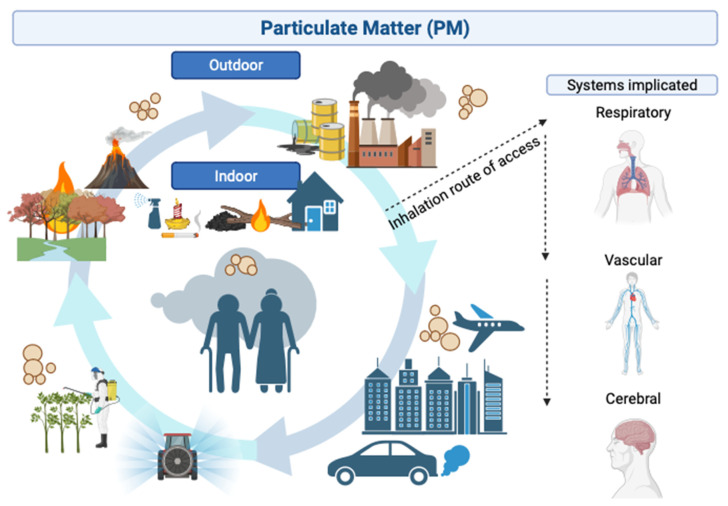
Overview of indoor and outdoor air pollution sources and their consequences to the human body. The main origin of outdoor PM is from a combination of industrial activity, internal combustion engines, traffic-related and geographic and meteorological conditions linked to increased forest fires and practices derived from agricultural activities. On the other hand, indoor environment particles originate from combustion, encompassing activities such as cooking and heating with coal, wood or dung, candles, incense, kerosene lamps, tobacco smoking, and non-combustion sources, such as cleaning and insecticide products, electric devices, and printers. PM enters the body via inhalation through the nose, increasing the risk of respiratory, cardiovascular, and cerebrovascular disease, therefore affecting the CNS. Figure created with BioRender.com (accessed on 10 October 2023).

**Figure 2 biomolecules-14-00035-f002:**
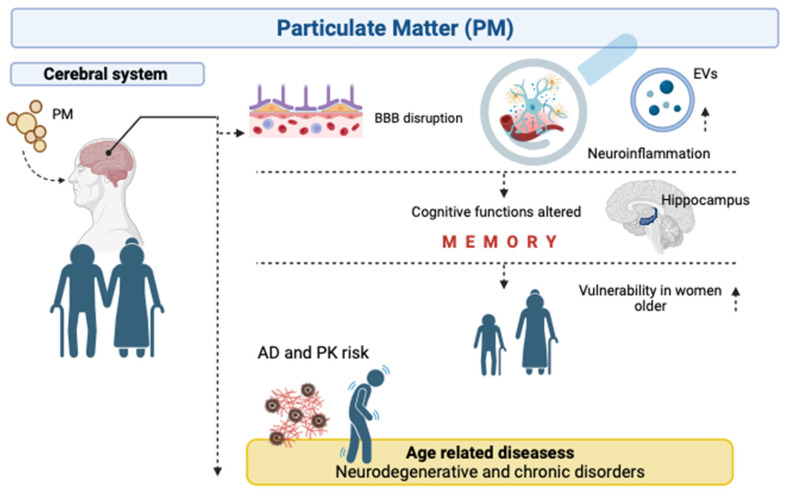
Overview of consequences in brain human by PM exposure. PM is capable of crossing blood–brain barrier and negatively affects the CNS by inflammation and EV deregulation. The result of such PM exposure is an impaired cognitive system during elderly, with higher vulnerability in women. Thus, it has been demonstrated an association between PM pollution and age-related diseases. Note: BBB, blood–brain barrier; PM, particulate matter; EVs, extracellular vesicles; AD, Alzheimer’s Disease and PK, Parkinson’s disease. Figure created with BioRender.com (accessed on 10 October 2023).

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
