# Peer review of "Particulate Matter in Human Elderly: Higher Susceptibility to Cognitive Decline and Age-Related Diseases"

_biomolecules, 2023, doi:10.3390/biom14010035_

Round 1

Reviewer 1 Report

Comments and Suggestions for Authors

The review manuscript by López-Granero and colleagues summarizes the current literature describing the risk of developing neurodegenerative disease as a result of particulate matter (PM) exposure. The work provides a well-organized and concise introduction to the generation and subsequent health effects of PM. The authors correctly highlight that there are very few epidemiological (or preclinical studies, for that matter) focusing on PM exposure in populations of advanced age. Therefore, I find the manuscript timely and important in addressing the paucity of data in this area. 

I have a few minor suggestions to improve the manuscript:

- Lines 72-73, add a reference for the statement regarding UFP/forest fires.
- Lines 201-203 could be worded more clearly. I believe the sentence is describing an increased risk per increase in PM exposure, but it's not completely clear as written.
- Lines 274-275 also could be worded more clearly.
- Throughout the manuscript, change instances of "gender" to "sex" for epidemiological studies and preclinical studies. Unless the epidemiological studies were specifically categorizing their population based on gender identity, sex is more appropriate in this case.
- Could the discussion on policy changes/public awareness be briefly expanded? It's highlighted in the abstract but there's only a very brief mention in the conclusion section. 

Author Response

Response to Reviewer 1 Comments

The review manuscript by López-Granero and colleagues summarizes the current literature describing the risk of developing neurodegenerative disease as a result of particulate matter (PM) exposure. The work provides a well-organized and concise introduction to the generation and subsequent health effects of PM. The authors correctly highlight that there are very few epidemiological (or preclinical studies, for that matter) focusing on PM exposure in populations of advanced age. Therefore, I find the manuscript timely and important in addressing the paucity of data in this area.

 I have a few minor suggestions to improve the manuscript:

  1. Lines 72-73, add a reference for the statement regarding UFP/forest fires.

Thanks, as suggested by the reviewer, we have included a new reference in line 73:

[9] Calderón-Garcidueñas, L.; Ayala, A. Air Pollution, Ultrafine Particles, and Your Brain: Are Combustion Nanoparticle Emissions and Engineered Nanoparticles Causing Preventable Fatal Neurodegenerative Diseases and Common Neuropsychiatric Outcomes? Environ Sci Technol 2022, 56, 6847–6856.

  1. Lines 201-203 could be worded more clearly. I believe the sentence is describing an increased risk per increase in PM exposure, but it's not completely clear as written.

We agree with the reviewer. We have modified these lines to better clarify the intent. The text reads as follow, see lines 203-206:

“In addition, 2.97 μg/m3 increase in PM2.5 exposure was related to poor memory and executive function. Furthermore, per 2.05 μg/m3 increase in PM2.5-10 exposure, global cognition, attention and verbal fluency declined. Also, per 4.94 μg/m3 increase in PM10 exposure, poor global cognition was noted in older adults [49]. “

  1. Lines 274-275 also could be worded more clearly.

We agree with the reviewer's comment and have addressed their suggestion. Now the reviewer can read the following in lines 277-280:

“Pro-inflammatory molecules were detected in blood as well as in cerebrospinal fluid after air pollution exposure [77]. The presence of these molecules causes brain damage by reducing neural tissue in specific brain areas, and by increasing the blood-brain-barrier permeability [78]“

  1. Throughout the manuscript, change instances of "gender" to "sex" for epidemiological studies and preclinical studies. Unless the epidemiological studies were specifically categorizing their population based on gender identity, sex is more appropriate in this case.

We thank to the reviewer for this suggestion. We have now changed “gender” to “sex” throughout the manuscript. See lines 118, 184, table 2 and 390.

  1. Could the discussion on policy changes/public awareness be briefly expanded? It's highlighted in the abstract but there's only a very brief mention in the conclusion section.

Thanks for this interesting comment. We have modified the discussion to include the policy changes/public awareness. The text in lines 515-532 has been changed as follows:

Finally, in line with Schraufnagel and collaborators [125], it is noteworthy that air pollution reduction is possible and the benefits of actions directed at such policies can result in substantial health gains, especially in risk populations such as elderly people. Several authors have even indicated that air pollution policies not only can improve human health, but are also relevant for decreasing climate changes. For example, Slovic and collaborators addressed how urban policies may improve air quality and mitigate global climate change [126]. Local initiatives and public awareness have largely focused on fuels and technology [126]. The benefits of the substitution of diesel taxis for alternatives fuels has also been addressed (such as hybrid, natural gas and liquefied petroleum gases) [127]. In Madrid, Spain, Vedrenne and collaborators examined the impact of Life Cycle Assessment (LCA) as instrument for policy-support [127], a means for studying the environmental impact of transport modes from cradle to grave [128]. The results showed that the shift to ecologic alternatives have positive impact of climate change and air pollution. Undoubtedly, any environmental policy will improve air quality, which translates into a healthier situation for the elderly living in cities. 

 Thus, changes in local policy setting agencies and governments along with greater awareness on the part of the general population will be the key for imparting a healthier natural environmental and reduced exposures to air contaminates.

Reviewer 2 Report

Comments and Suggestions for Authors

My suggestions:

1. I would add a table, that summarizes some examples of the particular matters, their origin, and which organs have a high risk of being affected. 

2. Instead of Figure 2, I would make two separate (and more detailed) figures (1) the role of PMs in neurodegeneration through inflammation (2) the role of PMs in neurodegeneration through extracellular vesicles.

3. Do PMs alter gene expressions, leading to neurodegeneration? The authors may discuss it.

4. Were PMs examined in other neurodegenerative diseases than AD and PD? For example multiple sclerosis, frontotemporal dementia, or ALS? If yes, the authors may mention it.

5. Is it possible that PMs could affect the mitochondrial functions?

Author Response

Response to Reviewer 2 Comments

My suggestions:

  1. I would add a table, that summarizes some examples of the particular matters, their origin, and which organs have a high risk of being affected.

We thank the reviewer for the suggestion. We have now included a new table. Table 1 have been introduced in line 161. Here is the table:

Table 1. Summaries of etiology of particulate matter and pathways implicated until reaching the CNS in the brain

PM by diameter

1 Main etiology

Deposited in:

References

PM0.1 or UFPM

≤ 100 nm

Smoke from wildfire

Cerebral cortex and cerebellum secondary to transport via the olfactory nerves

[9,26]

PM2.5

≤ 2.5 μm

Traffic-related activities, industrial sites and factories and agriculture

2 Lungs and is also a subject to olfactory transport and deposition in the olfactory cortex and other brain regions

[14,16,17,28]

PM10

≤ 10 μm

Pollen, mold, and dust particles

Filtered out by the

nose and upper airways

[14,16,17,27,28]

UFPM: ultrafine particles; CNS: central nervous system

Notes:

1 Outdoor environmental particles are primarily produced from road transport, agriculture, power plants, industry a and forest fires [9,14]. Indoor environmental particles are derived from combustion activities such as cooking, as well as and heating with coal, wood or dung, candles, incense, kerosene lamps, and tobacco smoking and non-combustion sources, and volatile organic compounds such as cleaning and insecticides products, electric de-vices and printers [2].

2 PM enters through the nose, once reaching the lung, PM may also travel through the blood into the brain [14,28].

  1. Instead of Figure 2, I would make two separate (and more detailed) figures (1) the role of PMs in neurodegeneration through inflammation (2) the role of PMs in neurodegeneration through extracellular vesicles.

We agree that it will be interesting to have 2 separate figures. However, there insufficient studies which address the role of VE in PMs neurodegeneration. We attempted to generate a figure, but it was deemed non-specific for the objective of this revision, since we were unable to determine the pathways of mechanics why VE are modified. Thus, we feel that the figure is incomplete and will not improve the quality of the manuscript.

Please, we ask for the reviewer to allow us keep the figure 2 as it has been presented in original version. Hope the reviewer accepts this decision.

  1. Do PMs alter gene expressions, leading to neurodegeneration? The authors may discuss it.

We agree with the reviewer’s comments. The gene expression alterations produced by air pollution and linked to human neurodegeneration, are discussed in lines 317-336.

4.3. Changes in gene expressions

The impact of air exposure has been linked to the development of neurodegenerative diseases, such as PD, associated with genetic alterations [83]. The molecular mechanisms of PD pathogenesis can be associated with epigenetic changes that alter gene expression without affecting the DNA sequence through DNA methylation, histone post-transcriptional modifications, and non-coding RNAs. However, environmental exposure can also cause alterations in cell function owing to changes in gene expression [83]. In PD, the most important epigenetic modifications are as follows: (1) DNA methylation (SNCA, PGC-1α, TNF- α, NOS2), (2) post-transcriptional histone methylation (H3K14, H3K9, and H3K18), and (3) microRNAs and other non-coding RNAs alterations [84,85]. Moreover, DNA methylation is related to other neurodegenerative disorders such as AD, depression, and Rett Syndrome. The most important genes for DNA methylation in AD are APP, BACE1, BIN1, and ANK1, as well as the co-occurring modifications to histones and expression of non-coding RNAs [86]. Furthermore, it is well established that PM have pro-oxidant activity damaging mitochondrial DNA that is more susceptible to oxidative injury and finally mitochondrial DNA gets methylated upon air pollution [81]. Lastly, an in vitro study in primary human neurons exposure to ultrafine PM has shown changes in non-coding RNAs implied in epigenetic regulation of gene expression and, therefore, alterations in gene expression related to neuroprotection such as a significant increase in the expression of metallothionein 1A and 1F with the propose to reduce PM damage in the brain [87].

  1. Were PMs examined in other neurodegenerative diseases than AD and PD? For example multiple sclerosis, frontotemporal dementia, or ALS? If yes, the authors may mention it.

From the available literature, we have provided a more detailed description on PM in other neurodegenerative diseases. Please see lines 398-417:

4.4.1 Others neurodegenerative diseases

There are few published studies on the relationship between PM and other neurodegenerative diseases like sclerosis (multiple or Amyotrophic Lateral Sclerosis [ALS]), or other types of dementia such as frontotemporal. Regarding multiple sclerosis, some authors have found no increased probability of developing multiple sclerosis after PM exposure [107]. On the contrary, in another study, using 9.072.576 participants, it was noted that PM10 concentrations explain 26,2% of the variance [108]. However, the multiple sclerosis cases in this article were self-reported, with imprecise air pollution measurements. For ALS, there is more evidence on this relationship. On the one hand, some works show no evidence of an increment in ALS risk after PM exposure [107,109,110]. On the other hand, other authors pointed out a slight increment in ALS prevalence [111] or visits to emergency departments for ALS symptoms [112]. In the case of other types of dementia, the published literature has not indicated an association linking PM2.5 and frontotemporal dementia [113].

Taken together, scientific evidence shows that air pollutants, specifically PM10 and PM2.5 is related to neurodegenerative disorders like AD, and PD. However, there is still insufficient information about the variables that may impact that relationship or the link between PM and others neurodegenerative diseases than AD and PD. Further research is needed to fully understand how the social environment, lifestyle, and other sociodemographic variables affect the relationship between air pollution and neurodegeneration. 

  1. Is it possible that PMs could affect the mitochondrial functions?

We are grateful for this suggestion. To date, few studies have shown a link between PM and mitochondrial dysfunction in human neurodegenerative diseases (also included in the answer to question #3). We have summarized the data in the following paragraph included in lines 305-315:

4.2. Mitochondrial function

Few studies have shown that mitochondria are targets in PM-induced human neurodegenerative disorders. PM can induce oxidative stress, and mitochondrial DNA has been shown to be affected due to altered DNA methylation; therefore, mitochondrial damage [81] secondary to air PM pollution is associated with the development of human cardiovascular and neurodegenerative diseases. Furthermore, PM may lead to dysregulation of neurohormonal stress pathways and trigger inflammation as well as oxidative stress affecting mitochondria. The mechanistic impact of PM-induced mitochondrial damage on human neurodegenerative diseases is incompletely understood [82]. Therefore, further investigations on mitochondrial dysfunction associated with human neurodegenerative diseases by PM are require understanding this complex pathogenesis.

Round 2

Reviewer 2 Report

Comments and Suggestions for Authors

The authors fulfilled my suggestions. Thank you.,